# The psychological impacts of the COVID-19 pandemic on business leadership

**Steven Mesquiti**[1]*, Sarah Seraj[2,3]

**1** Annenberg School for Communication, The University of Pennsylvania, Philadelphia, PA, United States of America, **2** Chief Technology Officer, A Better Force, Austin, TX, United States of America, **3** Department of Psychology, The University of Texas at Austin, Austin, TX, United States of America

* steven.mesquiti@asc.upenn.edu

## Abstract

The COVID-19 pandemic had a profound impact on business leadership, specifically on chief executive officers (CEOs). To document the psychological impacts of the pandemic on corporate leadership, this study analyzed the language of CEOs during company quarterly earnings calls ($N$ = 19,536) one year before and after the onset of the pandemic. Following the start of lockdowns, CEOs exhibited significant language shifts. Analytic thinking declined, and their language became less technical and more personal and intuitive. CEOs also showed signs of increased cognitive load as they grappled with the pandemic's impact on their business practices. The study observed a substantial decrease in collective-focused language (we-usage) among CEOs, indicative of disconnection from their companies. Concurrently, there was an increase in self-focused (I-usage) language, suggesting heightened preoccupation among business leaders. The observed language changes reflect the unique effect of the pandemic on CEOs, which had some notable differences compared to the general population. This study sheds light on how the COVID-19 pandemic influenced business leaders' psychological states and decision-making strategies—processes that have a substantial impact on a company's performance. The findings underscore the importance of language data in understanding large-scale societal events.

## Introduction

### The psychological impacts of the COVID-19 pandemic on corporate leadership

The 2019 coronavirus (COVID-19) pandemic severely altered the lives of people and companies across the globe. Over three years later, scientists have begun to document the vast psychological effects the pandemic has had on society [1–5]. Much of this work has relied on cross-sectional findings, with a growing number of studies beginning to document the pandemic's long-term effects [6]. Few studies have extended this work to organizational leadership, with no studies (to our knowledge) specifically examining *how* COVID affected the psychological states of business leaders. As a result, we know surprisingly little about the social and psychological effects the pandemic has had on the thinking of company leaders. However,

**Data Availability Statement:** The datasets "Big_CEO_copy.csv," "BLM_LIWC22_cleaned.csv," and "LIWC_BLMProject_22cities_01012016-04302021_weeklyavg.csv" are described in this paper. The raw text is not provided to preserve the

confidentiality of these discussions. Instead of the raw texts, the outputs from the text analysis tool LIWC are provided in the repository to replicate the study's findings. The raw text used in this project is available upon request to the authors. Data are available here: https://osf.io/sybea/?view_only= 5812674bd2ba4f618318bb94725ba443. The R scripts "Reddit-analyses.Rmd," "monthly-corr. Rmd," and "CEO-markdown.Rmd" can be used to replicate all of the analyses and figures in the main paper. The R script "SUPPLEMENTAL GRAPHS.R" can be used to reproduce all figures used Supporting Information materials. Additional statistical analyses from Supplementary Information is not provided in this repository but is available upon request.

**Funding:** The authors received no specific funding for this work.

**Competing interests:** The authors have declared that no competing interests exist

analyzing their communication in the form of quarterly earnings calls should provide a window into Chief Executive Officers' (CEOs) psychological shifts. This is especially important given how much influence a CEO's rationale and decision-making has on their company's performance, and in some cases the greater U.S. economy.

Research has shown that peoples' psychological states correlate heavily with their communication styles [7, 8]. For example, individuals have been found to use higher rates of first-person singular and lower rates of first-person plural pronouns when experiencing psychological distress during social and emotional upheavals [9], while lower usage of articles and prepositions have been linked to more dynamic styles of thinking [10]. With this information in mind, analyzing CEOs' communicated responses before, during, and after the start of the pandemic may offer insight into how their thinking and focus was affected.

Past studies that examined the effects of short-term upheavals (e.g., natural disasters) have highlighted changes in people's focus and cognitive processes [11, 12]. These findings produced social stage models of coping [13], which show how people navigate said upheavals [14, 15]. Social stage models of coping operate under the assumption that short-term upheavals often transpire within a few days, allowing people to gather themselves quickly after an upheaval and navigate the recovery period. In observing the psychological effects of short-term upheavals using social media data, researchers found that individuals discuss crisis-related topics during the event and for up to approximately two months after [13, 15]. Yet, the COVID pandemic has been anything but brief and predictable, as society still reels from the effects of irregular surges and novel variants three years later.

## Pandemics as social upheaval events

Unlike earthquakes or hurricanes, outbreaks of disease often lack an explicit date of conclusion, which can lead to heightened levels of negative emotions, like anxiety, for prolonged periods. Guntuku et al. [16] found that individuals reported elevated states of loneliness, stress, and anxiety in social media posts after the declaration of the COVID-19 pandemic as a global health emergency. Ashokkumar and Pennebaker [17] corroborate these findings using language as a proxy for psychological states and track the effects over time. In their examination of 18 U.S. city subreddits (communities on Reddit) and large-scale survey data, they observed distinct psychological shifts as the pandemic unfolded. At the pandemic's onset, there was an increase in people's use of anxiety-words and cognitive processing words (language people use when working through problems of limited understanding), as well as a decrease in analytical thinking (language that represents logical, hierarchical thinking). Nearly six weeks after the pandemic's onset, while large shifts in psychological states had stabilized, there continued to be shifts in these states around surges of the virus, remaining elevated relative to pre-pandemic levels. Given the severity of the pandemic, it is reasonable to assume that CEOs' language would display similar fluctuations, even if the magnitude of the effects vary compared to the general population.

## The language of CEOs

The ways in which CEOs communicate, as well as cope with upheavals, may be *different* from the general population [18], at least when it comes to their public discourse. Some studies suggest that CEOs are a unique set of individuals, such that successful CEOs have a propensity to score higher on measures of execution, charisma, and strategic focus [18]. However, work by Pfeffer [19] disputes these kinds of claims. They argue that these qualities do not accurately capture what leaders and CEOs are actually like, and that these assumptions arise from individuals' propensity to view leaders as infallible and omnipotent. These qualities may be

attributed to their roles necessitating the use of business rhetoric to mask sentiment in their language [20]. Instead of revealing their true thoughts and feelings in their communication with shareholders, CEOs may have to present a facade as a representative for their company. Nevertheless, some of their true feelings may be unconsciously revealed through their verbal communication. While people have the ability to control their use of content words, function word usage is much harder to manipulate [20]. Testing if CEOs experienced the pandemic in a similar or different way to the general population, through examining their language use, may settle some of these questions or otherwise yield useful insights.

**Analytic thinking and cognitive processing.** Dealing with the uncertainty of the pandemic should affect the ways that CEOs think and communicate their thoughts to analysts and shareholders via quarterly earnings calls. The increase in anxiety and uncertainty surrounding the success of their companies may tax their cognitive resources. Past work has found that when people experience stressful events such as the end of a romantic relationship or natural disaster, they expend cognitive effort to process the ordeal [9, 13, 15]. Often, people question *why* the event is happening, or *what* they can do to alleviate their distress. As a result of the taxing events of the pandemic, one would expect CEOs' cognitive styles to use more immediate, impulsive decision-making shortcuts [21], rather than organized, logical, and hierarchical thinking. Two language-based patterns—*analytic thinking* and *cognitive processing*—allow researchers to capture these cognitive styles through people's language use.

Analytic thinking involves formal, structured thought, which people use to understand and rationalize complicated problems. It may be interpreted as being calculated, where the person lays out a problem in a methodical manner. In the context of CEOs, analytic thinking is especially relevant during earnings calls given their structured format. In these calls, CEOs typically provide a statement, but also field unscripted questions from analysts and shareholders. Several measures of analytic thinking have been linked to written language assessment [22, 23]. In particular, Pennebaker et al. [24] employed factor analysis on the function words of college admissions essays, identifying a single factor where articles and prepositions were positively loaded and pronouns and negations were negatively loaded. Examples of texts with lower analytic scores include but are not limited to: speeches and communications of leaders who write and speak more informally [10, 25] and authors who write about more personal, emotional topics than more factual ones [26]. The development of the analytic thinking dimension shows that language analysis has the potential to accurately capture the thinking patterns of groups and individuals. Pertaining to business leaders, examining analytic thinking around the onset of the pandemic may elucidate how the pandemic affected their mental organization.

The second system, cognitive processing (or working through), occurs when people are actively trying to understand issues for which they have limited or no knowledge of. Markers of cognitive processing include insight words (e.g., understand, meaning), causal words (e.g., causes, result), and discrepancy words (e.g., would, should). Pennebaker et al. [27] suggests that cognitive processing words are often used at higher rates to detail negative events, as people attempt to produce a coherent narrative for why an event occurred. Increases in cognitive processing language have been linked to people in the midst of emotional upheavals [9, 28]. With this in mind, examining changes in CEOs' cognitive processing language around the onset of the pandemic has the potential to reveal how their active decision-making unfolded as they dealt with the crisis.

## Collective vs. Self-focused language

During stressful events, individuals often rely on introspection to inform themselves, leading to increases in stress, anxiety, and rumination [29]. An effective method for tracking an

individual's inward focus is by measuring their first-person singular pronouns or *self-focused language* usage. Studies have repeatedly found that elevated rates of self-focused language is associated with the experience of emotional upheavals, distress, and negative emotions [29–32]. Within the context of leadership research, higher rates of self-focused language have been associated with increased psychological distance and disconnection when used by people in positions of power [33]. Therefore, examining self-focused language within our period of interest may produce a reliable way of tracking CEOs' internal focus, distress, and preoccupation with their company and role as the pandemic unfolded.

In order for an organization with a social hierarchy to function effectively, there should be some sort of cohesion. Kacewicz et al. [34] and Chung and Pennebaker, [35] highlight these social hierarchies through the analysis of first-person plural pronouns or *collective-focused language*. Individuals of higher status use higher rates of collective language relative to their lower status peers, which is often related to increases in employee involvement and satisfaction [36, 37]. Conversely, leaders with lower usage can be perceived as uninvolved, cold, calloused, and disconnected [33, 38]. As a result, examining how collective-focused language varies may show how changes in CEOs' interconnectedness with their companies varied as a function of the pandemic.

**Tracking psychological states with text analysis.** In an age where human lives are immersed with technology and text data are readily available (e.g., audio transcripts, tweets, etc.), new methods for studying human behavior have emerged. The study of language using archival data has made it possible to observe the impact of large-scale events on people's thought processes and social behaviors on a level that was near impossible a few decades ago [39]. In the present study, we analyzed the quarterly earnings calls from over 2,300 publicly traded companies on the New York Stock Exchange between March 2019 and March 2021. Earnings calls are unique in that they can be used as proxies for CEOs' cognition, offering insight into their own thinking. In these calls, business analysts often present CEOs with questions that they are not fully prepared for. This prevents CEOs from relying on 'pre-baked' responses, providing relatively 'organic' speech patterns. As it is not possible to access the internal communications of most organizations, these calls provide a unique way to observe how CEOs communicate their decision-making to others. As a result, we pose the following research questions: (i) How did changes in CEO's attention and cognitive styles relate to the onset of the pandemic? (ii) How do these changes compare to past upheaval research? (iii) Do changes in CEOs' language follow that of the general public during our period of interest?

Comparing CEO language before the pandemic with language used during and after lockdown will allow us to estimate the relative magnitude of the pandemic's impacts. The following set of hypotheses are posed:

1. There should be a decrease in CEOs' analytic thinking immediately after the start of the pandemic, because they are more likely to be looking inward to grapple with their companies' challenges as the pandemic unfolds. That is, CEOs' thinking patterns will become more informal and unstructured (quantitatively the opposite of the analytic thinking dimension).

2. CEOs' thinking should become more dynamic and less organized as the pandemic unfolds, as they craft responses to the novel, complex obstacles presented by the pandemic. This increase in unstructured thought will be captured in CEOs' increased use of cognitive processing words.

3. CEOs' thinking will become more self-focused after the start of the pandemic. The pandemic was a stressful experience and likely led to increases in psychological distress and

distancing among CEOs, which can be captured by higher rates of first-person singular pronouns (i.e., self-focused language).

4. CEOs' thinking will become less collective-focused after the start of the pandemic. Specifically, we expect collective-focused language, as measured by rates of first-person plural pronouns, to decrease when the pandemic occurs and gradually increase in the months following the pandemic. These decreases may be indicative of disjointed decision-making, uncertainty, and social isolation caused by the pandemic.

## Methods

### Data & materials

**CEO dataset.** To assess the effect the pandemic had on CEOs, the current study relied on archival data from a sample of company quarterly earnings calls, which are sessions required by the U.S. Securities and Exchange Commission. Using the Finnhub API, all unscripted earnings calls for companies on the New York Stock Exchange between January of 2006 and March of 2021 were extracted ($N = 79,725$ transcripts; $N = 4,707$ CEOs). Due to the low volume and temporal resolution of data before 2010, we excluded data from 2006–9 in our analyses.

To answer our main research questions, we constructed a sub-sample of all companies in the database ranging from March 2019 to March 2021 to acquire one year of data before and after lockdowns in the US. Transcripts of less than 25 words were excluded from analyses, which is a common cutoff for bag-of-words text analysis approaches like the one our team used ($N = 191$ transcripts excluded). The final sample was composed of 19,536 transcripts from 3,044 CEOs. Although it is difficult to obtain precise information about the demographics of individual CEOs, aggregated demographic information from previous research [40] suggests that CEOs are largely upper middle class, white males, in middle to late adulthood [41].

**Reddit dataset.** To understand how the pandemic impacted the psychological processes of the general population in comparison with CEOs, we relied on the social news and discussion website Reddit. Around 12 million unique users visit Reddit daily to participate in forums (subreddits) related to their interests and hobbies [40]. Reddit contains city-related subreddits, where people can talk about issues pertaining to that specific city. The current study extracted submissions and comments from US city subreddits that were posted between January 2016 and April 2021. We selected subreddits with at least 50,000 subscribers and were geographically dispersed across the US. This extracted dataset consisted of 33.7 million posts from 1.37 million users across 22 US city subreddits: r/Atlanta, r/Austin, r/Baltimore, r/boston, r/Charlotte, r/Chicago, r/Columbus, r/Dallas, r/Houston, r/LosAngeles, r/Minneapolis, r/nashville, r/newyorkcity, r/nyc, r/NewOrleans, r/Philadelphia, r/phoenix, r/Portland, r/sandiego, r/SeattleWA, r/StLouis, r/tampa, r/twincities, r/washingtondc. From the extracted dataset, we constructed a sub-sample of all posts ranging from March 2019 to March 2021 to compare shifts in the Redditors' language to that of CEOs. Similar to the CEO dataset, posts of less than 25 words were excluded from analyses [$N = 16,681,591$ posts excluded] to reduce noise when analyzing the data. The final sample was composed of 10,738,506 posts from 833,172 redditors. Reddit data was obtained using the Pushshift application programming interface [42]. Although it is difficult to obtain precise information about the demographics of individual reddit users, aggregated demographic information from previous surveys suggests Reddit users are 60–70% male with a mean age of 22–34 years old [40].

## Text analysis procedure

The earnings call transcripts were primarily analyzed using the text analysis program LIWC-22 (Linguistic Inquiry and Word Count 2022 [43]. LIWC is an automated, dictionary-based, text analysis program that calculates the percentage of words in a text that pertains to different social, psychological, and linguistic categories. The goal of the study was to measure CEOs' changes in language in the 12 months before and after the start of the pandemic. The analysis focuses on the following LIWC categories: analytic thinking, cognitive processing words, self-focused language (I-words), and collective-focused language (we-words). For text samples high on each dimension please see S1 File.

Analytic thinking is a standardized measure in LIWC, ranging from 0 to 100, with higher scores indicating higher levels of analytic thinking. Samples higher in analytic thinking possess higher rates of articles and prepositions, indicating references to objects, ideas, and linkages between them [24]. It is also related to more formal, structured, and impersonal language, which reflect logical, formal, and hierarchical thinking [25]. Texts with lower scores in analytic thinking have higher base rates of pronouns, adverbs, auxiliary verbs, negations, and conjunctions, which is associated with more here-and-now, narrative thinking [26].

Unlike analytic thinking, which is a standardized measure, the remaining dimensions indicate the percentage of total words in a text corresponding to that category. Cognitive processing includes words related to conceptualizing and understanding events (e.g., think, because, perhaps), and reflect attempts to understand or work through complex problems. Self-focus is measured by the usage of first-person singular pronouns. Collective-focus is measured through the use of we-words (i.e., first person plural pronouns). While only four dimensions from LIWC-22 are presented in main analyses, results from additional dimensions of interest are provided in the S1 File.

## Results

### Reference point for the pandemic

The linguistic patterns of CEOs were examined for up to a year before and after the onset of the COVID-19 pandemic in March of 2020. A reference point of March 2019 was established as a comparison time point for each hypothesis to capture pre-COVID language trends 12 months before the pandemic lockdowns started in the US. CEOs' did not actively begin discussing COVID until January of 2020, and it eventually became a prominent discussion topic by March of that year. Fig 1 depicts graphical trends for language categories corresponding to analytic thinking, cognitive processes, collective-focus, and self-focus. Means for each language dimension were calculated by month to capture fluctuations in their language use as the pandemic rapidly unfolded (see S1 File for details on analytic strategy). To analyze language shifts in CEOs, Welch's paired t-tests were conducted to compare the established reference point of March 2019 against each month in the period of interest. Welch's paired t-tests were used to account for an unequal number of calls in each month, as earnings calls can occur at any point throughout a fiscal quarter. Each month after lockdowns started was also compared to the previous month to test whether there were significant month-to-month changes. Please see the project's OSF site: https://osf.io/sybea/?view_only=5812674bd2ba4f618318bb94725ba443 for de-identified data and analysis scripts.

**Analytic thinking.** As seen in Fig 1, CEOs' language became the most unstructured and informal immediately following lockdown in March 2020. Levels of analytic thinking largely remained below pre-pandemic levels in the subsequent months, consistent with our prediction that there would be decreases in analytic thinking after the onset of the pandemic. Significant

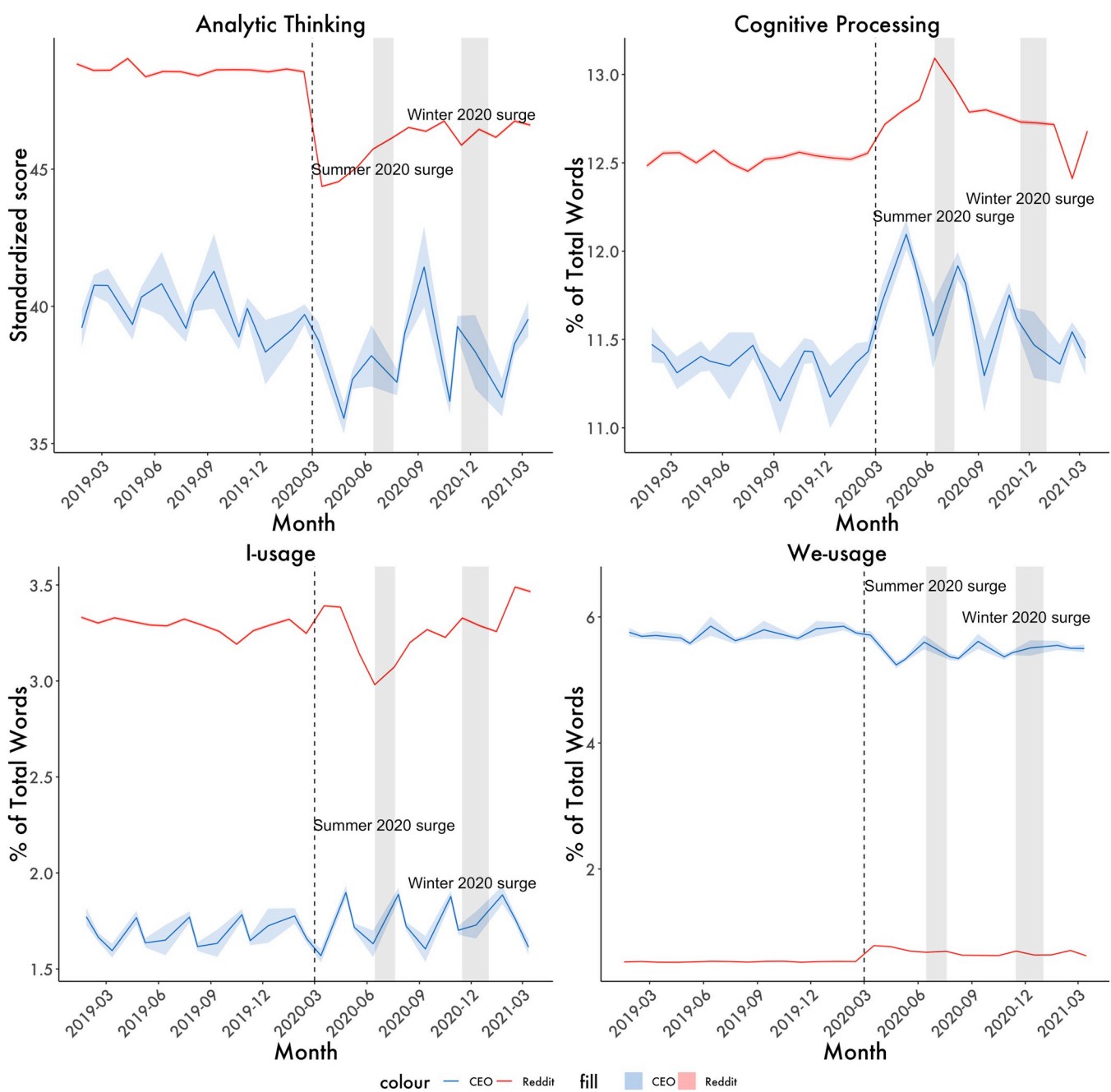

**Fig 1. CEOs' language change.** Change in language patterns of Reddit users and CEOs before and after the pandemic. The red line represents Reddit post history in the displayed time range ($N$ = 10,738,506 posts). The blue line refers to all CEO transcripts in the displayed time range ($N$ = 19,536 calls). Vertical Line Represents the onset of the pandemic. Each point is the average score across CEOS ($N$ = 3,044) Reddit users ($N$ = 834,172) at that time block (1 month intervals). The horizontal shaded areas are +/- 1 standard error for each datapoint, vertical shading represent virus surges. The self-focused language is measured using I-words and collective-focused using we-words.

decreases in analytic thinking, relative to the baseline, emerged in April [$t(623.94)$ = 5.777, $p < 0.001$, d = 0.395], the month after lockdowns began. Month-to-month decreases in analytic thinking after lockdown persisted into August of 2020 [$t(200.57)$ = -2.212, $p < 0.05$, d = -0.162] and began to stabilize by September 2020 [$t(218.93)$ = -1.687, $p$ = 0.093, d = -0.148;

hereafter only Cohen's d are reported for the sake of brevity, see S1 File for full results]. However, after several months of stability, levels of analytic thinking decreased again in January of 2021 (d = 0.268), following the COVID surge of Winter 2020.

**Cognitive processing.** There were increases in CEOs' usage of cognitive processing after the onset of the pandemic. Increases in cognitive processing words relative to the baseline emerged in March of 2020, as lockdown occurred (d = -0.107). Month-to-month increases persisted into July, where they were strongest (d = -0.320). Cognitive processing usage began to decrease in September of 2020 (d = 0.344), but remained elevated relative to baseline levels until the end of our dataset, never returning to pre-pandemic levels. The observed changes in cognitive processing were consistent with our prediction: rates of cognitive processing words in CEOs' language would increase after lockdown.

**Self-focused language.** Increases in CEOs' self-focused language–relative to the baseline–emerged after lockdown and became statistically significant in April of 2020 (d = -0.356; see Fig 1). This finding offers support for our prediction that CEOs would use higher rates of self-focused language after the start of the pandemic. In terms of monthly changes, CEOs' use of self-focused language decreased in May (d = 0.190) and June (d = 0.200) of 2020. As captured in Fig 1, changes in self-focused language continued to fluctuate in a cyclical manner over the next several months until the end of the dataset period, but stayed above March 2019 levels (see Fig 1). These results were consistent with our third hypothesis, as CEOs' use of self-focused language increased after the start of the pandemic.

**Collective-focused language.** CEOs' usage of collective-focused language decreased relative to the baseline after the pandemic began, becoming statistically significant in April (d = 0.223; see Fig 1). In observing monthly changes, collective-focused language began to increase in June (d = -0.259), yet decreased again in August (d = 0.250), with the strongest decreases in collective language occurring in October (d = 0.292). Month-to-month changes in CEO collective-focused language were no longer significant after November 2020 (d = -0.013), yet continued to remain below baseline levels in the subsequent months (see Fig 1). The observed changes in collective-focused language supported the fourth hypothesis, which was that collective language usage would decrease when the pandemic first occurred.

**Do changes in CEOs' language follow that of the general public?.** Changes in the language of CEOs appeared to follow that of the Reddit data across the cognitive language dimensions ($r_{Analytic}$ = .60, $p < .001$; $r_{cogproc}$ = 0.57, $p = 0.002$), which captures how CEOs and the general population followed similar trends in their thinking across our period of interest. Like that of the CEOs, Reddit users' language became the most unstructured and informal immediately following lockdown in April 2020 (d = 0.146), also failing to return to pre-pandemic levels in the subsequent months. There were also increases in Reddit users' usage of cognitive processing after lockdown in (d = -0.031), which remained above baseline levels similar to the CEOs. While overall use of self-focused language between CEOs and Redditors was unrelated (r = -0.12, $p = 0.57$), increases in Redditors' use of self-focused language also emerged in April 2020 (d = -0.025; see Fig 1), same as that of CEOs. Interestingly, there were stark differences between the Redditors and CEOs regarding collective-focused language use (r = - 0.68, $p < .001$). Unlike CEOs, Redditors experienced an increased use of collective-focused language after the pandemic began in April (d = -0.156). This was a marked difference from CEOs, who observed strong decreases across the period of interest (see Fig 1). The observed discrepancies in collective-focused language capture the unprecedented effects that the pandemic had on the fabric of business culture. At a time where most people were considering their personal well-being as well as that of their close friends and family, business leaders were additionally grappling with the fate of their companies in a completely new economic climate. Yet, this seemed

to have caused them to decrease in their collective-focused thinking, unlike what happened with the general public.

### Post hoc question

**Psychological impacts of COVID-19 compared to across the decade.** The results of our primary analyses raised an additional line of inquiry. That is, how did changes in CEOs' language during the pandemic compare to the language changes of the past decade? Were there disruptive business events in the past that could have caused language changes of similar magnitude to that experienced during the pandemic? To answer these questions, a larger range of dates was considered: 79,068 transcripts for 4,906 CEOs from January of 2010 to March of 2021. The resulting dataset contained 48 quarterly means ranging from 2010 to 2021. Fig 2 depicts the quarterly means for the focal language dimensions of interest. Quarterly means were calculated to avoid overweighting any particular month due to the irregular scheduling of calls earlier in the decade. In comparison to the average observed effects of events from Q1 of 2010 to Q4 of 2019, the effects observed at the start of the pandemic (Q1 2020) were substantially greater across the analytic thinking, cognitive processing, and collective focus dimensions. However, these shifts were not as strong when examining the self-focused language dimension. This may be the product of analysts consistently pressing CEOs to answer questions from their own, unique perspective, rather than that of the company's. Collectively, these findings show that the shifts in CEOs' thinking and focus, fueled by the pandemic, were unlike anything business leaders had encountered in recent history. This also applies to the general public, who experienced language and psychological shifts unlike anything in the last five years, as others have reported before (see Ashokkumar & Pennebaker 2021 for a more detailed analysis of the language shifts experienced by the general public due to the pandemic).

## Discussion

Beyond its physical and mental health impacts, the pandemic has had profound psychological implications for how CEOs organize and interact with their worlds. Utilizing real-world language data, we illustrate the pandemic's effects on CEOs' natural cognitive patterns and attention allocation and how these effects compare to a more general sample. Similar to past language research that has studied the psychological effects of the pandemic on the general public [e.g., 16, 17, 44], psychological disruptions emerged immediately following the onset of the pandemic. Contrasting CEOs' language during the period of interest with the previous decade suggests that the psychological impact of COVID-19 on business leaders was greater than anything the business world has experienced in recent history.

### Cognitive systems

Consistent with our first hypothesis, we observed decreases in analytic thinking in April of 2020 immediately following lockdown. This has value in capturing *how* this event severely affected the cognitive equilibrium of CEOs as they communicated with shareholders and analysts. For example, previous studies linked lower levels of analytic thinking to poorer academic performance and less hierarchical trains of thought [24, 25]. Decreases in CEOs' analytic thinking suggests they were unable to organize their world in a structured manner and that their decision-making process was inhibited. Our results extend past psycholinguistic work on the effect of upheavals [9, 44, 45] to that of CEOs, as both CEOs and Reddit users experienced large decreases in analytic thinking following lockdown.

CEOs' use of cognitive processing words were also affected, increasing after the onset of the pandemic, consistent with our second hypothesis and past work [9, 44, 45]. While changes in

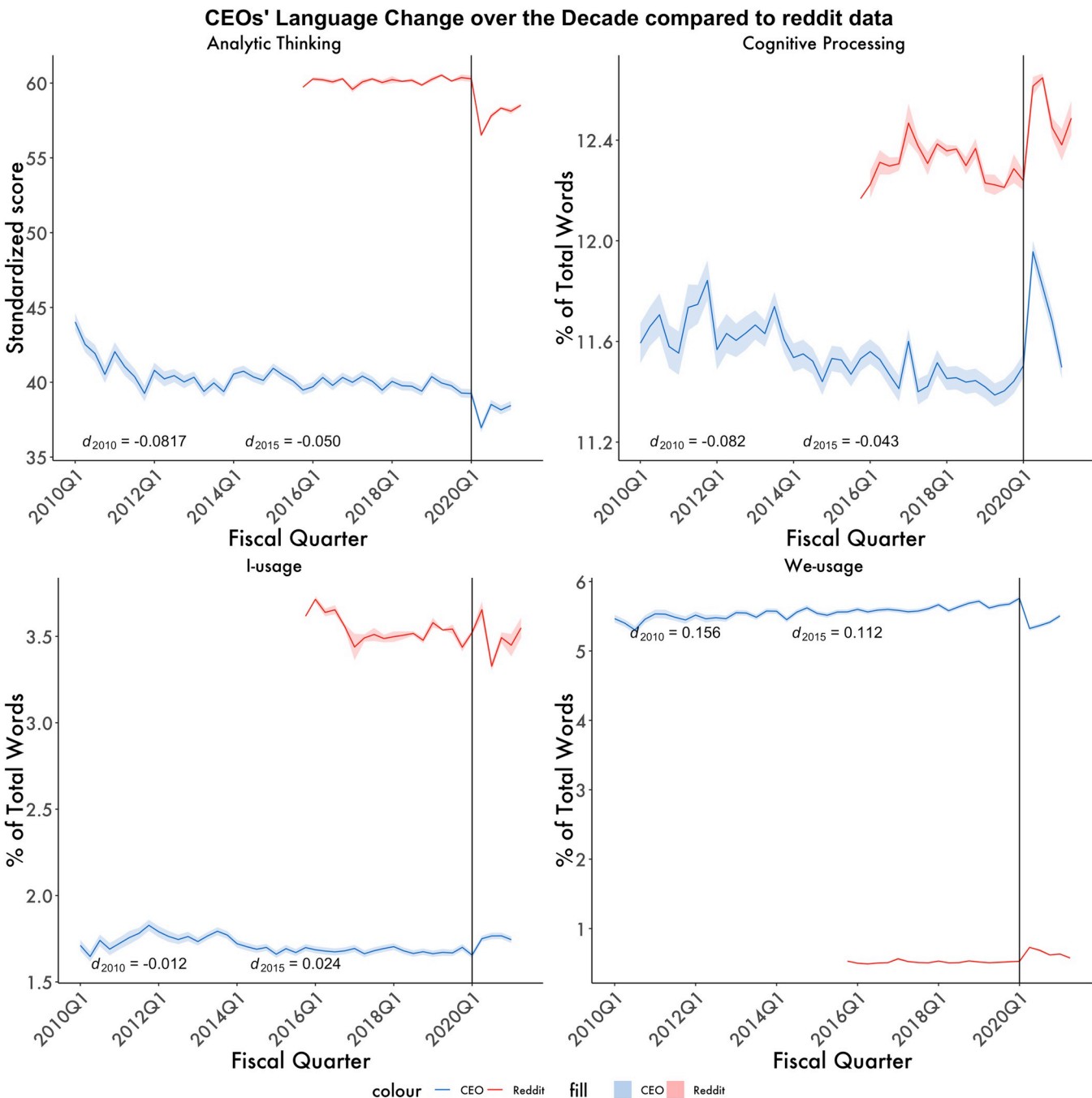

**Fig 2. CEOs' language change over the decade.** Change in language patterns of Reddit users and CEOs before and after the pandemic. The red line represents Reddit post history in the displayed time range ($N$ = 33.7 million posts). The blue line refers to all CEO transcripts in the displayed time range ($N$ = 79,725 transcripts). The vertical Line Represents the onset of the pandemic. Each point is the average score across CEOs ($N$ = 4,707 CEO) and Redditors ($N$ = 1.37 million) at that time block (1 quarter intervals; 3 months). The horizontal shaded areas are 95% CIs for each datapoint, vertical shading represent virus surges. The self-focused language is measured using I-words and collective-focused using we-words. Effect sizes presented are the average effect size at 5 and 10 years for the CEO dataset.

cognitive processing language levels stabilized after October 2020, they remained elevated relative to pre-pandemic levels—following a similar pattern to that of Reddit users. These shifts highlight the prolonged cognitive load that executives were experiencing, similar to that of the

general US population, whose language showed similar shifts. The increases in cognitive processing language suggest that while CEOs were able to deal with the initial shock of the pandemic and lockdown, they still struggled to adapt to the persistent challenges COVID presented even up to a year later.

Taken together, the analyses of cognitive language display shifts in the ways CEOs were thinking as the pandemic unfolded. The fluctuation in analytic and cognitive processing words reveal two dynamic cognitive systems that CEOs used to navigate the pandemic. This sheds light on how CEOs process novel, stressful events and actively work through them to produce viable solutions for their companies, as well as how these methods were inherently different than that of the average person. For example, cognitive processing increased more for CEOs than the general population, which captures the high stress nature of their jobs during this period. There was also a smaller reduction in CEOs' analytic thinking compared to the general public, which captures how CEO language was not as unstructured in our period of interest—a function of the required formality in earnings calls. Our findings reinforce how CEOs continued to grapple with the effects of the pandemic up to a year later, as levels of analytic thinking and cognitive processing did not return to pre-pandemic baseline levels by the end of our dataset. Unlike Redditors, CEOs cannot just worry about themselves and their immediate family and friends; they must also consider the financial well-being of their company *and* stakeholders. Tracking these individuals' language may help to reveal subconscious disruptions in their thinking during stressful situations, which can often be masked in their rhetoric. The disruption of thought patterns are illustrated by the following excerpts from earnings calls presented in Table 1.

## Attentional focus

In addition to changes in CEOs' thinking, their attentional focus also shifted, as seen in their use of first-person pronouns. We observed large reductions in collective-focused language, particularly in October 2020, and this remained below pre-pandemic levels for a year, aligning with our fourth hypothesis. The decline in collective-focused language suggests that CEOs' decision-making and rationalization were detached from their company and employees

**Table 1. LIWC dimensions of interest example.**

| LIWC Dimension | Example Text | |
|---|---|---|
| Analytic Thinking | Example 1 | "...I think **as** *we* sit here *we* feel **like** *we have* really good results **from** operations *and* feel **like** every quarter *it's* **going to** *be* **a** very positive EBITDA quarter every quarter *of* 2020. *So that* said, obviously COVID-19 issue is evolving almost every hour *or* so..." |
| | Example 2 | "*I mean*, *we* saw **the** same *thing* back *when* **the** financial crisis **of** 2008 occurred, it turned **out to** be **a** blip **for us**..." |
| Cognitive Processing | Example 1 | "...I **think** as we sit here we **feel** like we have **really** good **results** from operations and **feel** like every quarter it's going to be a **very positive** EBITDA quarter every quarter of 2020. So that said, **obviously** COVID-19 issue is evolving **almost** every hour **or** so..." |
| | Example 2 | "**I mean**, we saw the **same** thing back when the financial crisis of 2008 occurred, it turned out to be a blip for us..." |

*Note*. Words in bold result in higher scores for the respective LIWC Dimension. Analytic thinking is the only LIWC dimension presented that is a bidirectional dimension. The words in bold would cause analytic thinking scores to be higher, with the words in italics causing the scores to be lower. For cognitive processing, higher scores indicate that a greater % of words pertaining to that category was present in the sample.

during this period. Previous research offers support for this finding, as decreased use of collective-focused language by leaders has been linked to lack of involvement and disconnection [33, 34], as well as lower overall employee satisfaction and engagement [37]. The decrease in collective-focus language was inconsistent with that of the Reddit dataset, which experienced an increase post-March 2020. This discrepancy, perhaps, underscores the stark differences in attentional shifts between CEOs and that of the general public, while also capturing how unprepared business leaders were for the unprecedented effects of the pandemic in the spring and summer of 2020.

We also noticed a rise in CEO self-focused language compared to the baseline immediately after the pandemic began, similar to that of the general public. Increases in self-focused language were observed in April, July, and October 2020, with the largest increase in February 2021, aligning with our third hypothesis. These increases may indicate that CEOs experienced high levels of stress and a sense of detachment from their company during these periods when the virus was spreading rapidly. Our finding is consistent with previous studies [31, 32], which linked higher rates of self-focused language to increased distress and negative emotions. In the context of leadership research, greater usage of self-focused language has been associated with psychological distance, lack of warmth, and disconnection [33]. However, it's important to note that the changes in CEOs' self-focused language appeared to follow a cyclical trend and were rather noisy relative to that of the Reddit dataset. Observed changes in self-focused language also occurred every other month or so, differing from the other three language variables we examined. The observed fluctuations may suggest that factors other than the pandemic influenced these shifts, or may have been the product of the quarterly earnings call format, which lacks the temporal resolution of data from large social media platforms (e.g., Twitter).

Collectively, examining shifts in first-person pronouns aided in our understanding of CEOs' attentional shifts as a function of a global pandemic. Indeed, changes in these two attention types help to capture the level of discord that the pandemic incited in the organizational leadership of companies, as well as some distinct qualities relative to the general public. Our findings indicate the pandemic may have not unified the business world, unlike other forms of upheaval [46, 47]. In fact, the pandemic appears to have divided businesses and their leadership, captured by large decreases in collective focus language post-pandemic, which also did not return to baseline in our period of interest. These large shifts may have been the product of the uncertainty and novelty surrounding the pandemic. In examining self-focus language, there were increases relative to baseline immediately after the pandemic began, yet these trends became rather cyclical post-lockdown. The large increases in self-focus language revealed CEOs' potential high levels of stress and a sense of detachment from their company when the novel-virus was first spreading rapidly. Tracking how these individuals' attention changes when experiencing stressful events, via language analysis, may help scientists to better understand how CEOs' internal focus and distress changes as they experience these events.

## Post hoc findings

To date, most upheaval research has focused on specific events that had major impacts on a general group of people over a relatively brief time. Our analysis of CEOs' language across the decade suggests that the pandemic had a unique effect on how the business leaders of the world carried out their responsibilities. Our findings also demonstrate that the pandemic was unlike anything company leaders have encountered between 2011–2021. The closest example of an equivalent crisis for business leaders was the financial crisis of 2007–8, but unfortunately there was not enough CEO transcript data dating back to that time to do a proper comparison. Arguably, the pandemic is an upheaval that surpasses the magnitude of the financial crisis of

2007–8, given its widespread impacts on the general population, beyond just financial matters and how it completely altered societal functioning. As a result, CEOs have been playing catch-up since March of 2020, attempting to actively solve problems for an event for which there is no script or road map. Taken with our main findings, our post hoc findings demonstrate the value in using language data to understand large-scale societal events. In the future, similar techniques may permit scientists to better understand executive leaderships' cognition as 'once in a lifetime' events become more common.

## Limitations, and future directions

The present study has a number of strengths including a strong theoretical base, utilization of valid language analysis techniques, and high external validity due to the use of large-scale social media data. However, it is important to note that this study was not without its limitations. The temporal analysis focused on large publicly-traded companies. As such, it is not informative regarding the psychological effects of COVID on smaller or privately-traded businesses. Additionally, it is important to note the period of interest in the study overlaps with some other large-scale national events such as the Black Lives Matter protests in the Summer of 2020 following the killing of George Floyd, the 2020 US presidential election, and the January 6th Capitol riots. While the observed effects in our study appear strongest with surges of the virus, it is important to acknowledge the presence of other events—and their potential effects—in the period of interest. While using LIWC allows for the exploration of psychological states via text analysis, it is not without its limitations. As a closed vocabulary approach, its internal dictionaries are not exhaustive. LIWC also does not consider the context in which words are used [48] unlike large language models [LLM] such as Bidirectional Encoder Representations from Transformers [commonly referred to as BERT; 49] or Generative Pre-trained Transformer [the LLM developed by OpenAI that powers the chatbot ChatGPT; 50]. While these LLMs provide more accurate results when it comes to context-specific settings, they are computationally intensive and harder to interpret compared to the LIWC method, which is more accessible. Lastly, this study largely examined the psychological impacts on the CEOs without probing individual differences (e.g., race and gender) due to the lack of demographic information. Since past research has shown gender differences in language use [51] as well as leadership style [52], future research should explore individual differences in the observed psychological effects.

While the current study provides insight regarding the psychological effects of COVID on a novel sample, there are a number of future research avenues that may be particularly valuable. One such avenue is testing whether the observed effects replicate when companies are split into their respective sectors. Vetting sector-level effects would be especially interesting given that different sectors experienced different magnitudes of financial hardships when the economy crashed in March of 2020. In addition, testing for gender differences in CEOs language may be worthwhile. There are unique gender differences in language [51], such that women tend to use more words related to psycho-social processes. On the other hand, men display a tendency to orient themselves to object properties and impersonal topics. Gender difference exploration should be coupled with the growing body of literature exploring gender differences in leadership styles and leadership styles [52, 53]. These two extensions may one day help to capture individual differences in the effective rhetoric of business leaders. Lastly, we encourage future researchers to test the relationship between a company's CEO and their market value on the Stock Exchange. Identifying this relationship has the potential to help researchers better understand the degree to which CEOs' oration is driven by their companies' financial standing on the open market.

The use of open-sourced data on platforms such as Finnhub, Reddit, Twitter, and Facebook enables researchers to observe human behavior using large sample sizes. Our study of almost 5,000 CEOs across 2,300 companies (supplemented with 10.7 million Reddit posts from over 833,000 redditors across 22 U.S. subreddits) highlight the merit of using language analysis to gain insights on leaders' decision-making, as well as how these trends compare to that of the general public. Within weeks of crises unfolding, it was possible to identify large shifts in business leaders' thinking at the time of crisis, something never before accomplished on this scale. Our design also allowed us to identify the length of time that these disruptive changes lasted, on average lasting for 7 months. For example, increases in cognitive processing language and decreases in analytic thinking post-lockdown highlight the increased cognitive load and distribution in structured thought that CEOs were experiencing. There is a growing body of literature on the potential disruptions that cognitive load and reduced working memory have on decision-making, work performance, and other important cognitive outcomes [54]. A key contribution of our work is that it underscores the power of using language data to see how collective upheavals, like the pandemic, affect business leadership. More importantly, it allows researchers to track how upheavals influence the ways that CEOs think and make decisions, as well as the effect these changes may have on the performance of their companies and the greater global economy.

## Supporting information

**S1 File. Supporting information.**
(DOCX)

## Acknowledgments

We thank James Pennebaker and the Pennebaker Lab, Kate Blackburn, Mia Carranza, Danielle Cosme, and Emily Falk for their feedback on earlier versions of the manuscript. Additionally, we thank Michael Durland for his assistance with procuring the data.

## Author Contributions

**Conceptualization:** Steven Mesquiti.

**Data curation:** Steven Mesquiti.

**Formal analysis:** Steven Mesquiti, Sarah Seraj.

**Investigation:** Steven Mesquiti, Sarah Seraj.

**Methodology:** Steven Mesquiti, Sarah Seraj.

**Supervision:** Sarah Seraj.

**Visualization:** Steven Mesquiti.

**Writing – original draft:** Steven Mesquiti, Sarah Seraj.

**Writing – review & editing:** Steven Mesquiti, Sarah Seraj.

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
