## [Decision Letter · Decision Letter 0]

2 May 2023

PONE-D-22-32858The Psychological Impacts of the COVID-19 Pandemic on Corporate LeadershipPLOS ONE

Dear Dr. Mesquiti,

Thank you for submitting your manuscript to PLOS ONE. After careful consideration, we feel that it has merit but does not fully meet PLOS ONE’s publication criteria as it currently stands. Therefore, we invite you to submit a revised version of the manuscript that addresses the points raised during the review process.

We look forward to receiving your revised manuscript.

Kind regards,

Grace Akello, Ph.D

Academic Editor

PLOS ONE

Additional Editor Comments:

Dear Author,

We now reviews for your article and although they both agree that your work is important they have several points regarding how to improve your article. The abstract, background and findings need to be more succinct and resonate with the title. And if your focus is Post Covid, please find sufficient information showing status-quo data from the same CEOs. Doing this will enable us appreciate the impact on their management characteristics before COVID-19.

Additionally, it will be important to revise the entire text, so that it reads more like an article rather than a summary of a dissertation thesis.

Regards

Grace Akello, PhD

Reviewers' comments:

Reviewer's Responses to Questions

**Comments to the Author**

1. Is the manuscript technically sound, and do the data support the conclusions?

Reviewer #1: Yes

Reviewer #2: Yes

2. Has the statistical analysis been performed appropriately and rigorously? 

Reviewer #1: Yes

Reviewer #2: Yes

3. Have the authors made all data underlying the findings in their manuscript fully available?

Reviewer #1: Yes

Reviewer #2: Yes

4. Is the manuscript presented in an intelligible fashion and written in standard English?

Reviewer #1: Yes

Reviewer #2: Yes

5. Review Comments to the Author

Reviewer #1: The manuscript provides an interesting aspect of the pandemic impacts and offers unique insights on how economic leaders' psychology shifted during the pandemic. The analysis is done based on the sound theoretical framework. My major concern at the sane time is the focus on CEOs. While this is an interesting angle and motivated well in the introduction, the findings should be able to highlight some unique aspects of CEOs. That being said, the manuscript should provide some comparisons to general public or other cohorts. Else, the focus on CEOs cannot be justified. Any perspective on this should be addressed in the discussion section using prior literature or in the limitation section.

Further, some editing as well as restructuring seem necessary. Currently, the manuscript reads more like a short version of thesis or dissertation. The introduction should be shorter and more succinct. The first paragraph under Method is not a method. There should be another section for Data and Material (or alike).

Overall, I feel that this work is worthy of publication after addressing the points raised above.

Reviewer #2: This manuscript addresses an interesting and important issue in psychological research: The Effects of COVID-19 Pandemic on CEO Language Change. The literature can benefit from studies that clarify the mood induction procedures.

My comments are organized according to the sections of the paper, with the most important issues in each section addressed first. The following are some concerns and recommendations for strengthening this paper.

Title: “The Effects of COVID-19 Pandemic on CEO Language Change” is more appropriate and accurate as a title than “The Psychological Impacts of the COVID-19 Pandemic on Corporate Leadership” Perhaps, authors could mix both titles.

Abstract: It would be useful to include a broad description of the conclusions in the abstract.

Keywords. COVID, Pandemic and Leadership keywords are redundant with the title. Perhaps, authors could include another keywords.

Perhaps “Linguistic Inquiry and Word Count” is more appropriate as keyword than LIWC.

Introduction

The introduction should include more aspects that describes the reasons why interest in the study Linguistic Inquiry and Word Count approach has increased (or the benefits of its increase) and what gaps this research intends to cover on Corporate Leadership analyses.

Methods

Authors follow a systematic approach. The raw sample extracted from Finnhub possessed 79,725 transcripts from 4,707 CEOs. Could authors analyze other sociodemographic variables (i.e. age, social class) or corporate characteristics (i.e. primary secondary or tertiary sector companies)? I suggest including these variables as covariates in the analyses.

Please also create a first table presenting the characteristics of the sample.

Authors state on page10 that “Extremely short texts of less than 25 words were omitted”. Short texts are common in open-sourced data on platforms. Why authors excluded them?

How many researchers are resolving the conflicts (i.e. when they prepared text samples for LIWC analysis)?

Authors state on page 9 that “Using the Finnhub API, all of the unscripted earnings calls for companies on the New York Stock Exchange between January of 2006 and March of 2021 were extracted

Nevertheless they stated on page 10 that “The raw sample extracted from Finnhub possessed 79,725 transcripts from 4,707 CEOs. From that, a sample of all companies in the database ranging from March 2019 to March 2021 was constructed, to acquire one year of data before and after lockdowns in the US.

Have the analysed 2006-2021 data or from 2019?

Results

Authors state on page 12 that “To analyze language shifts in CEOs, Welch's paired t-tests were conducted to compare the established reference point of March 2019 against each month in the period of interest.”

Welch's t-test is generally applied when the there is a difference between the variations of two populations and also when their sample sizes are unequal. What criteria was used to select this statistical analysis?

Discussion

Authors recognized some limitations on page 20. Nevertheless, Do the characteristics of the program used for the analysis allow it to capture the non-literal meaning of the expressions or the influence of the context on what is expressed? It would be useful to explain the limitations of the Linguistic Inquiry and Word Count 22 program.

Bibliography

PLOS uses “Vancouver” style. Please include the Journal name abbreviation in the following references:

2. Fitzpatrick KM, Harris C, Drawve G. Fear of COVID-19 and the mental health consequences in America. Psychological Trauma: Theory, Research, Practice, and Policy. 2020 Aug;12(S1):S17–21. doi:10.1037/tra0000924.

4. Shaw P, Blizzard S, Shastri G, Kundzicz P, Curtis B, Ungar L, Koehly L. A daily diary study into the effects on mental health of COVID-19 pandemic-related behaviors. Psychological Medicine. 2021 Jul 12:1-9.

5. Kang Y, Cosme D, Pei R, Pandey P, Carreras-Tartak J, Falk EB. Purpose in life, loneliness, and protective health behaviors during the COVID-19 pandemic. The Gerontologist. 2021;61(6):878–87.

6. Luchetti M, Lee JH, Aschwanden D, Sesker A, Strickhouser JE, Terracciano A, et al. The trajectory of loneliness in response to COVID-19. American Psychologist. 2020 Oct;75(7):897–908. doi:10.1037/amp0000690.

7. Tausczik YR, Pennebaker JW. The Psychological Meaning of Words: LIWC and Computerized Text Analysis Methods. Journal of Language and Social Psychology. 2010 Mar;29(1):24–54. doi:10.1177/0261927X09351676.

8. Mitchell MA, Capron DW, Raines AM, S 504 chmidt NB. Reduction of cognitive concerns of anxiety sensitivity is uniquely associated with reduction of PTSD and depressive symptoms: A comparison of civilians and veterans. Journal of Psychiatric Research. 2014;48(1):25–31.

11. Nolen-Hoeksema S, Morrow J. A prospective study of depression and posttraumatic stress symptoms after a natural disaster: the 1989 Loma Prieta Earthquake. Journal of personality and social psychology. 1991;61(1):115.

12. Silver RC, Holman EA, McIntosh DN, Poulin M, Gil-Rivas V. Nationwide longitudinal study of psychological responses to September 11. Jama. 2002;288(10):1235–44.

13. Pennebaker JW, Harber KD. A Social Stage Model of Collective Coping: The Loma Prieta Earthquake and The Persian Gulf War. Journal of Social Issues. 1993 Jan;49(4):125–45. doi:10.1111/j.1540-4560.1993.tb01184.x.

14. Jones NM, Silver RC. This is not a drill: Anxiety on Twitter following the 2018 Hawaii false missile alert. American Psychologist. 2020;75(5):683. 15. Guntuku SC, Sherman G, Stokes DC, Agarwal AK, Seltzer E, Merchant RM, et al. Tracking mental health and symptom mentions on Twitter during COVID-19. Journal of general internal medicine. 2020;35(9):2798–800.

17. Ashokkumar A, Pennebaker JW. Social media conversations reveal large psychological shifts caused by COVID-19’s onset across US cities. Science advances. 2021;7(39):eabg7843.

18. Kaplan SN, Sorensen M. Are CEOs Different? The Journal of Finance. 2021;76(4):1773–811.

22. Schmader T, Johns M. Converging evidence that stereotype threat reduces working memory capacity. Journal of Personality and Social Psychology. 2003;85(3):440–52. doi:10.1037/0022-3514.85.3.440.

23. Kleim B, Horn AB, Kraehenmann R, Mehl MR, Ehlers A. Early Linguistic Markers of Trauma-Specific Processing Predict Post-trauma Adjustment. Frontiers in Psychiatry [Internet]. 2018;9. Available from: https://www.frontiersin.org/article/10.3389/fpsyt.2018.00645

25. Mehrabian A, Wiener M. Non-immediacy between communicator and object of communication in a verbal message: application to the inference of attitudes. Journal of Consulting Psychology. 1966;30(5):420.

28. Jordan KN, Sterling J, Pennebaker JW, Boyd RL. Examining long-term trends in politics and culture through language of political leaders and cultural institutions. Proceedings of the National Academy of Sciences. 2019;116(9):3476–81.

29. Boyd RL, Blackburn KG, Pennebaker JW. The narrative arc: Revealing core narrative structures through text analysis. Science advances. 2020;6(32):eaba2196.

31. Chung C, Pennebaker JW. The psychological functions of function words. Social communication. 2007;1:343–59.

33. Tackman AM, Sbarra DA, Carey AL, Donnellan MB, Horn AB, Holtzman NS, Edwards TM, Pennebaker JW, Mehl MR. Depression, negative emotionality, and self-referential language: A multi-lab, multi-measure, and multi-language-task research synthesis. Journal of personality and social psychology. 2019 May;116(5):817.

34. Rude S, Gortner EM, Pennebaker J. 572 Language use of depressed and depression

vulnerable college students. Cognition & Emotion. 2004;18(8):1121–33.

35. Slatcher RB, Chung CK, Pennebaker JW, Stone LD. Winning words: Individual differences in linguistic style among U.S. presidential and vice presidential candidates. Journal of Research in Personality. 2007 Feb;41(1):63–75. doi:10.1016/j.jrp.2006.01.006.

36. Kacewicz E, Pennebaker JW, Davis M, Jeon M, Graesser AC. Pronoun Use Reflects Standings in Social Hierarchies. Journal of Language and Social Psychology. 2014 Mar;33(2):125–43. doi:10.1177/0261927X13502654.

38. Weiss M, Kolbe M, Grote G, Spahn DR, Grande B. We can do it! Inclusive leader language promotes voice behavior in multi-professional teams. The Leadership Quarterly. 2018;29(3):389–402.

39. Mayfield J, Mayfield M. Leader‐level influence on motivating language: A two‐level model investigation on worker performance and job satisfaction. Competitiveness Review: An International Business Journal. 2010,

40. Pennebaker JW, Slatcher RB, Chung CK. Linguistic markers of psychological state through media interviews: John Kerry and John Edwards in 2004, Al Gore in 2000. Analyses of Social Issues and Public Policy. 2005;5(1):197–204

41. Giorgi S, Nguyen KL, Eichstaedt JC, Kern ML, Yaden DB, Kosinski M, et al. Regional personality assessment through social media language. Journal of personality.

2022;90(3):405–25.

44. Niederhoffer K, Mooth R, Wiesenfeld D, Gordon J. The origin and impact of CPG new product buzz: Emerging trends and implications. Journal of Advertising Research. 2007 Dec 1;47(4):420-6.

45. Markowitz DM. Psychological trauma and emotional upheaval as revealed in academic writing: The case of COVID-19. Cognition and Emotion. 2022;36(1):9–22.

46. Cohn MA, Mehl MR, Pennebaker JW. Linguistic markers of psychological change surrounding September 11, 2001. Psychological science. 2004;15(10):687–93.

47. Deck C, Jahedi S, Sheremeta R. On the consistency of cognitive load. European Economic Review. 2021;134:103695.

48. Hawdon J, Ryan J. Social relations that generate and sustain solidarity after a mass tragedy. Social forces. 2011;89(4):1363–84.

49. Hawdon J, Ryan J, Agnich L. Crime as a source of solidarity: a research note testing Durkheim’s assertion. Deviant Behavior. 2010;31(8):679–703.

50. Kern ML, Park G, Eichstaedt JC, Schwartz HA, Sap M, Smith LK, et al. Gaining insights from social media language: Methodologies and challenges. Psychological methods. 2016;21(4):507.

51. Matz SC, Gladstone JJ, Stillwell D. In a 617 world of big data, small effects can still matter: A reply to Boyce, Daly, Hounkpatin, and Wood (2017). Psychological science. 2017;28(4):547–50.

52. Newman ML, Groom CJ, Handelman LD, Pennebaker JW. Gender differences in language use: An analysis of 14,000 text samples. Discourse processes. 2008;45(3):211– 36.

53. Oakley JG. Gender-based barriers to senior management positions: Understanding the scarcity of female CEOs. Journal of business ethics. 2000;27(4):321–34.

54. Zhang G, Jia Z, Yan S. Does Gender Matter? The Relationship Comparison of Strategic Leadership on Organizational Ambidextrous Behavior between Male and Female CEOs. Sustainability. 2022;14(14):8559.

Recommendation: accept with minor changes, because despite these limitations, few previous studies have explored the Effects of COVID-19 Pandemic on CEO Language Change.

6. PLOS authors have the option to publish the peer review history of their article (what does this mean?). If published, this will include your full peer review and any attached files.

Reviewer #1: **Yes: **Naoru Koizumi

Reviewer #2: No

---

## [Author Response · Author response to Decision Letter 0]

9 Jul 2023

Editor Prompts: 

There are no ethical or legal restrictions on sharing the de-identified dataset. 

The anonymized datasets are available at the following link: : https://osf.io/sybea/?view_only=5812674bd2ba4f618318bb94725ba443 for the anonymized datasets necessary to replicate our study.

Reviewer #1: 

The manuscript provides an interesting aspect of the pandemic impacts and offers unique insights on how economic leaders’ psychology shifted during the pandemic. The analysis is done based on the sound theoretical framework. My major concern at the sane time is the focus on CEOs. While this is an interesting angle and motivated well in the introduction, the findings should be able to highlight some unique aspects of CEOs. That being said, the manuscript should provide some comparisons to general public or other cohorts. Else, the focus on CEOs cannot be justified. Any perspective on this should be addressed in the discussion section using prior literature or in the limitation section.

Thank you for your comments. We agree that contextualizing our findings with that of a more general population is rather important. As a result, we have included some comparisons to the general public by analyzing and adding Reddit data to our graphs (10,738,506 comments from 834,172 users in 22 US city subreddits in the period from March 2019 to March 2021) . Please see Methods section for more information on data collection). Analyses with the additional data is now incorporated into the Results and Discussion as well as Figures 1 and 2 (see below). We have also highlighted some of the unique aspects of CEOs in comparison with the general public. 

Fig 1. CEOs' Language Change. Change in language patterns of Reddit users and CEOs before and after the pandemic. The red line represents Reddit post history in the displayed time range (N = 10,738,506 posts). The blue line refers to all CEO transcripts in the displayed time range (N = 19,536 calls).Vertical Line Represents the onset of the pandemic. Each point is the average score across CEOS (N = 3,044) Reddit users (N = 834,172) at that time block (1 month intervals). The horizontal shaded areas are 95% CIs for each datapoint, vertical sharing represent virus surges. The self-focused language is measured using I-words and collective-focused using we-words.

Fig 2. CEOs' Language Change over the Decade. Change in language patterns of Reddit users and CEOs before and after the pandemic. The red line represents Reddit post history in the displayed time range (N = 33.7 million posts). The blue line refers to all CEO transcripts in the displayed time range (N = 79,725 transcripts). The vertical Line Represents the onset of the pandemic. Each point is the average score across CEOs (N = 4,707 CEO) and Redditors (N = 1.37 million) at that time block (1 quarter intervals; 3 months). The horizontal shaded areas are 95% CIs for each datapoint, vertical sharing represent virus surges. The self-focused language is measured using I-words and collective-focused using we-words. Effect sizes presented are the average effect size at 5 and 10 years for the CEO dataset. 

Further, some editing as well as restructuring seem necessary. Currently, the manuscript reads more like a short version of thesis or dissertation. The introduction should be shorter and more succinct. The first paragraph under Method is not a method. There should be another section for Data and Material (or alike).

● Thank you for your constructive comments, which have greatly strengthened the manuscript. We have restructured the paper and condensed some of the sections. As per your feedback, we also have a separate Data Section. 

Reviewer #2: 

Title: “The Effects of COVID-19 Pandemic on CEO Language Change” is more appropriate and accurate as a title than “The Psychological Impacts of the COVID-19 Pandemic on Corporate Leadership” Perhaps, authors could mix both titles.

We appreciate your insight here. We have switched the title to: “The Psychological Impacts of the COVID-19 Pandemic on Business Leadership” to better capture the nature of the paper. 

Abstract: It would be useful to include a broad description of the conclusions in the abstract.

Thank you for the feedback. We have updated the abstract to add the big picture conclusions, which are reproduced below:

The observed language changes reflect the unique effect of the pandemic on CEOs, which had some notable differences compared to the general population. This study sheds light on how the COVID-19 pandemic influenced business leaders' psychological states and decision-making strategies—processes that have a substantial impact on a company’s performance. The findings underscore the importance of language data in understanding large-scale societal events.

Keywords. COVID, Pandemic and Leadership keywords are redundant with the title. Perhaps, authors could include another keywords.Perhaps “Linguistic Inquiry and Word Count” is more appropriate as a keyword than LIWC.

● We appreciate your feedback here. The updated keywords are now: LIWC, COVID, Language, CEOs, Social Upheaval. Since LIWC is widely used as an acronym for Linguistic Inquiry and Word Count, we still kept “LIWC” as a keyword in the spirit of parsimony.

Introduction

The introduction should include more aspects that describes the reasons why interest in the study Linguistic Inquiry and Word Count approach has increased (or the benefits of its increase) and what gaps this research intends to cover on Corporate Leadership analyses.

We appreciate your feedback. We now touch on how past social stage models of coping might mirror that of CEOs, as well as how CEOs might differ from lay people on page 5. Also on page 5, we provide empirical evidence that establishes some of the unique interpersonal qualities of CEOs, as well as how language analysis (specifically the analysis of function words) might allow us to test whether these unique qualities were present during the pandemic.

Methods

Authors follow a systematic approach. The raw sample extracted from Finnhub possessed 79,725 transcripts from 4,707 CEOs. Could authors analyze other sociodemographic variables (i.e. age, social class) or corporate characteristics (i.e. primary secondary or tertiary sector companies)? I suggest including these variables as covariates in the analyses.Please also create a first table presenting the characteristics of the sample.

Thank you very much for your clarifying question. Unfortunately, these variables are not readily available in this dataset due to it only possessing the names and not ethnicity nor gender. However, we provide some general information regarding the demographic qualities of CEOs on page 10:

”Although it is difficult to obtain precise information about the demographics of individual CEOs, aggregated demographic information suggest that CEOs are largely upper middle class, white males, in middle to late adulthood [Larcker & Tayan, 2020]...”

Authors state on page10 that “Extremely short texts of less than 25 words were omitted”. Short texts are common in open-sourced data on platforms. Why did the authors excluded them?

Thank you for raising this point. We have addressed this comment on page 10: 

Transcripts of less than 25 words were excluded from analyses, which is a common cutoff for bag-of-words text analysis approaches like the one our team used (N = 191 transcripts excluded). 

Essentially, Short transcripts would cause output to be excessively noisy, since LIWC’s output calculates the percentage of total words that pertain to different categories. Moreover, the 25 word cutoff only excludes a small percentage of texts in our dataset, usually ones that don’t have much information to begin with since they are quick replies to questions instead of a more detailed answer.

How many researchers are resolving the conflicts (i.e. when they prepared text samples for LIWC analysis)?

The earning calls transcripts are part of a well-maintained database through Finnhub, which aims to democratize financial data and make it more accessible to analysts. We did not have to do any additional cleaning of the text samples to process it through LIWC. As such, there weren’t any conflicts to resolve. 

Authors state on page 9 that “Using the Finnhub API, all of the unscripted earnings calls for companies on the New York Stock Exchange between January of 2006 and March of 2021 were extracted. Nevertheless they stated on page 10 that “The raw sample extracted from Finnhub possessed 79,725 transcripts from 4,707 CEOs. From that, a sample of all companies in the database ranging from March 2019 to March 2021 was constructed, to acquire one year of data before and after lockdowns in the US.Have the analysed 2006-2021 data or from 2019?

Thank you for your clarifying question. We obtained all data from 2006-2021 from the Finnhub API. However, we only analyzed data from 2010-2021 due to there being a scarce number of observations before 2010. This point is now made more clear in the Methods section on page 10: 

“Using the Finnhub API, all unscripted earnings calls for companies on the New York Stock Exchange between January of 2006 and March of 2021 were extracted (N = 79,725 transcripts; N = 4,707 CEOs). Due to the low volume and temporal resolution of data before 2010, we excluded data from 2006-9 in our analyses.

Results

Authors state on page 12 that “To analyze language shifts in CEOs, Welch’s paired t-tests were conducted to compare the established reference point of March 2019 against each month in the period of interest.”Welch’s t-test is generally applied when the there is a difference between the variations of two populations and also when their sample sizes are unequal. What criteria was used to select this statistical analysis?

We used Welch’s paired t-tests to account for an unequal number of calls in each month, as these calls can occur at any point in a fiscal quarter. We have added the following information to page 13: “

Welch's paired t-tests were conducted to compare the established reference point of March 2019 against each month in the period of interest. We used Welch’s paired t-tests to account for an unequal number of calls in each month.” 

Discussion

Authors recognized some limitations on page 20. Nevertheless, Do the characteristics of the program used for the analysis allow it to capture the non-literal meaning of the expressions or the influence of the context on what is expressed? It would be useful to explain the limitations of the Linguistic Inquiry and Word Count 22 program.

Thank you for your clarifying question. We have added the following information to page 26:

“While using LIWC allows for the exploration of psychological states via text analysis, it is not without its limitations. As a closed vocabulary approach, its internal dictionaries are not exhaustive. LIWC also does not take into account the context in which words are used [Schwartz et al., 2013] unlike large language models [LLM] such as Bidirectional Encoder Representations from Transformers [commonly referred to as BERT; Devlin et al., 2018] or Generative Pre-trained Transformer [the LLM developed by OpenAI that powers the chatbot ChatGPT; Brown et al., 2023]. While these LLMs provide more accurate results when it comes to context-specific settings, they are computationally intensive and harder to interpret compared to the LIWC method, which is more accessible.”

---

## [Editor Report · Decision Letter 1]

14 Aug 2023

The Psychological Impacts of the COVID-19 Pandemic on Business Leadership

PONE-D-22-32858R1

Dear Mr. Mesquiti

We’re pleased to inform you that your manuscript has been judged scientifically suitable for publication and will be formally accepted for publication once it meets all outstanding technical requirements.

Kind regards,

Grace Akello, Ph.D

Academic Editor

PLOS ONE
---

## [Editor Report · Acceptance letter]

18 Sep 2023

PONE-D-22-32858R1 

The Psychological Impacts of the COVID-19 Pandemic on Business Leadership 

Dear Dr. Mesquiti:

I'm pleased to inform you that your manuscript has been deemed suitable for publication in PLOS ONE. Congratulations! Your manuscript is now with our production department. 

Kind regards, 

on behalf of

Dr. Grace Akello 

Academic Editor

PLOS ONE